# Safe Policy Improvement by Minimizing Robust Baseline Regret

**Marek Petrik**
University of New Hampshire
mpetrik@cs.unh.edu

**Mohammad Ghavamzadeh**
Adobe Research & INRIA Lille
ghavamza@adobe.com

**Yinlam Chow**
Stanford University
ychow@stanford.edu

## Abstract

An important problem in sequential decision-making under uncertainty is to use limited data to compute a *safe* policy, which is guaranteed to outperform a given baseline strategy. In this paper, we develop and analyze a new *model-based* approach that computes a safe policy, given an inaccurate model of the system's dynamics and guarantees on the accuracy of this model. The new robust method uses this model to directly minimize the (negative) regret w.r.t. the baseline policy. Contrary to existing approaches, minimizing the regret allows one to improve the baseline policy in states with accurate dynamics and to seamlessly fall back to the baseline policy, otherwise. We show that our formulation is NP-hard and propose a simple approximate algorithm. Our empirical results on several domains further show that even the simple approximate algorithm can outperform standard approaches.

## 1 Introduction

Many problems in science and engineering can be formulated as a sequential decision-making problem under uncertainty. A common scenario in such problems that occurs in many different fields, such as online marketing, inventory control, health informatics, and computational finance, is to find a good or an optimal strategy/policy, given a batch of data generated by the current strategy of the company (hospital, investor). Although there are many techniques to find a good policy given a batch of data, only a few of them guarantee that the obtained policy will perform well, when it is deployed. Since deploying an untested policy can be risky for the business, the product (hospital, investment) manager does not usually allow it to happen, unless we provide her/him with some performance guarantees of the obtained strategy, in comparison to the baseline policy (for example the policy that is currently in use).

In this paper, we focus on the *model-based* approach to this fundamental problem in the context of *infinite-horizon* discounted Markov decision processes (MDPs). In this approach, we use the batch of data and build a *model* or a *simulator* that approximates the true behavior of the dynamical system, together with an *error function* that captures the accuracy of the model at each state of the system. Our goal is to compute a *safe* policy, i.e., a policy that is guaranteed to perform at least as well as the baseline strategy, using the simulator and error function. Most of the work on this topic has been in the *model-free* setting, where *safe* policies are computed directly from the batch of data, without building an explicit model of the system [Thomas *et al.*, 2015b,a]. Another class of *model-free* algorithms are those that use a batch of data generated by the current policy and return a policy that is guaranteed to perform better. They optimize for the policy by repeating this process until convergence [Kakade and Langford, 2002; Pirotta *et al.*, 2013].

A major limitation of the existing methods for computing safe policies is that they either adopt a newly learned policy with provable improvements or do not make any improvement at all by returning the baseline policy. These approaches may be quite limiting when model uncertainties are not uniform

across the state space. In such cases, it is desirable to guarantee an improvement over the baseline policy by combining it with a learned policy on a state-by-state basis. In other words, we want to use the learned policy at the states in which either the improvement is significant or the model uncertainty (error function) is small, and to use the baseline policy everywhere else. However, computing a learned policy that can be effectively combined with a baseline policy is non-trivial due to the complex effects of policy changes in an MDP. Our key insight is that this goal can be achieved by minimizing the (negative) *robust regret* w.r.t. the baseline policy. This unifies the sources of uncertainties in the learned and baseline policies and allows a more systematic performance comparison. Note that our approach differs significantly from the standard one, which compares a pessimistic performance estimate of the learned policy with an optimistic estimate of the baseline strategy. That may result in rejecting a learned policy with a performance (slightly) better than the baseline, simply due to the discrepancy between the pessimistic and optimistic evaluations.

The *model-based* approach of this paper builds on *robust* Markov decision processes [Iyengar, 2005; Wiesemann *et al.*, 2013; Ahmed and Varakantham, 2013]. The main difference is the availability of the baseline policy that creates unique challenges for sequential optimization. To the best of our knowledge, such challenges have not yet been fully investigated in the literature. A possible solution is to solve the robust formulation of the problem and then accept the resulted policy only if its conservative performance estimate is better than the baseline. While a similar idea has been investigated in the *model-free* setting (e.g., [Thomas *et al.*, 2015a]), we show in this paper that it can be overly conservative.

As the main contribution of the paper, we propose and analyze a new robust optimization formulation that captures the above intuition of minimizing robust regret w.r.t. the baseline policy. After a preliminary discussion in Section 2, we formally describe our model and analyze its main properties in Section 3. We show that in solving this optimization problem, we may have to go beyond the standard space of deterministic policies and search in the space of randomized policies; we derive a bound on the performance loss of its solutions; and we prove that solving this problem is NP-hard. We also propose a simple and practical approximate algorithm. Then, in Section 4, we show that the standard model-based approach is really a tractable approximation of robust baseline regret minimization. Finally, our experimental results in Section 5 indicate that even the simple approximate algorithm significantly outperforms the standard model-based approach when the model is uncertain.

## 2 Preliminaries

We consider problems in which the agent's interaction with the environment is modeled as an *infinite-horizon* $\gamma$-discounted MDP. A $\gamma$-discounted MDP is a tuple $\mathcal{M} = \langle \mathcal{X}, \mathcal{A}, r, P, p_0, \gamma \rangle$, where $\mathcal{X}$ and $\mathcal{A}$ are the state and action spaces, $r(x, a) \in [-R_{\max}, R_{\max}]$ is the bounded reward function, $P(\cdot|x, a)$ is the transition probability function, $p_0(\cdot)$ is the initial state distribution, and $\gamma \in (0, 1]$ is a discount factor. We use $\Pi_R = \{\pi : \mathcal{X} \rightarrow \Delta^{\mathcal{A}}\}$ and $\Pi_D = \{\pi : \mathcal{X} \rightarrow \mathcal{A}\}$ to denote the sets of *randomized* and *deterministic* stationary Markovian policies, respectively, where $\Delta^{\mathcal{A}}$ is the set of probability distributions over the action space $\mathcal{A}$.

Throughout the paper, we assume that the true reward $r$ of the MDP is known, but the true transition probability is not given. The generalization to include reward estimation is straightforward and is omitted for the sake of brevity. We use historical data to build a MDP *model* with the transition probability denoted by $\widehat{P}$. Due to limited number of samples and other modeling issues, it is unlikely that $\widehat{P}$ matches the true transition probability of the system $P^{\star}$. We also require that the estimated model $\widehat{P}$ deviates from the true transition probability $P^{\star}$ as stated in the following assumption:

**Assumption 1.** *For each* $(x, a) \in \mathcal{X} \times \mathcal{A}$*, the error function* $e(x, a)$ *bounds the* $\ell_1$ *difference between the estimated transition probability and true transition probability, i.e.,*

$$\|P^{\star}(\cdot|x, a) - \widehat{P}(\cdot|x, a)\|_1 \le e(x, a). \tag{1}$$

The error function $e$ can be derived either directly from samples using high probability concentration bounds, as we briefly outline in Appendix A, or based on specific domain properties.

To model the uncertainty in the transition probability, we adopt the notion of robust MDP (RMDP) [Iyengar, 2005; Nilim and El Ghaoui, 2005; Wiesemann *et al.*, 2013], i.e., an extension of

MDP in which nature adversarially chooses the transitions from a given *uncertainty set*

$$\Xi(\widehat{P}, e) = \left\{ \xi : \mathcal{X} \times \mathcal{A} \to \Delta^{\mathcal{X}} : \|\xi(\cdot|x, a) - \widehat{P}(\cdot|x, a)\|_1 \le e(x, a), \ \forall x, a \in \mathcal{X} \times \mathcal{A} \right\}.$$

From Assumption 1, we notice that the true transition probability is in the set of uncertain transition probabilities, i.e., $P^\star \in \Xi(\widehat{P}, e)$. The above $\ell_1$ constraint is common in the RMDP literature (e.g., [Iyengar, 2005; Wiesemann *et al.*, 2013; Petrik and Subramanian, 2014]). The uncertainty set $\Xi$ in RMDP is $(x, a)$-rectangular and randomized [Le Tallec, 2007; Wiesemann *et al.*, 2013]. One of the motivations for considering $(x, a)$-rectangular sets in RMDP is that they lead to tractable solutions in the conventional reward maximization setting. However, in the robust regret minimization problem that we propose in this paper, even if we assume that the uncertainty set is $(x, a)$-rectangular, it does not guarantee tractability of the solution. While it is of great interest to investigate the structure of uncertainty sets that lead to tractable algorithms in robust regret minimization, it is beyond the main scope of this paper and we leave it as future work.

For each policy $\pi \in \Pi_R$ and nature's choice $\xi \in \Xi$, the discounted *return* is defined as

$$\rho(\pi, \xi) = \lim_{T \to \infty} \mathbb{E}_\xi \left[ \sum_{t=0}^{T-1} \gamma^t r(X_t, A_t) \mid X_0 \sim p_0, \ A_t \sim \pi(X_t) \right] = p_0^\top v_\pi^\xi,$$

where $X_t$ and $A_t$ are the state and action random variables at time $t$, and $v_\pi^\xi$ is the corresponding *value function*. An *optimal policy* for a given $\xi$ is defined as $\pi_\xi^\star \in \arg\max_{\pi \in \Pi_R} \rho(\pi, \xi)$. Similarly, under the true transition probability $P^\star$, the *true return* of a policy $\pi$ and a *truly optimal policy* are defined as $\rho(\pi, P^\star)$ and $\pi^\star \in \arg\max_{\pi \in \Pi_R} \rho(\pi, P^\star)$, respectively. Although we define the optimal policy using $\arg\max_{\pi \in \Pi_R}$, it is known that every reward maximization problem in MDPs has at least one optimal policy in $\Pi_D$.

Finally, given a deterministic *baseline* policy $\pi_B$, we call a policy $\pi$ *safe*, if its "true" performance is guaranteed to be no worse than that of the baseline policy, i.e., $\rho(\pi, P^\star) \ge \rho(\pi_B, P^\star)$.

## 3 Robust Policy Improvement Model

In this section, we introduce and analyze an optimization procedure that robustly improves over a given baseline policy $\pi_B$. As described above, the main idea is to find a policy that is guaranteed to be an improvement for any realization of the uncertain model parameters. The following definition formalizes this intuition.

**Definition 2** (The Robust Policy Improvement Problem). *Given a model uncertainty set $\Xi(\widehat{P}, e)$ and a baseline policy $\pi_B$, find a maximal $\zeta \ge 0$ such that there exists a policy $\pi \in \Pi_R$ for which $\rho(\pi, \xi) \ge \rho(\pi_B, \xi) + \zeta$, for every $\xi \in \Xi(\widehat{P}, e)$.*[1]

The problem posed in Definition 2 readily translates to the following optimization problem:

$$\pi_S \in \arg\max_{\pi \in \Pi_R} \min_{\xi \in \Xi} \left( \rho(\pi, \xi) - \rho(\pi_B, \xi) \right). \tag{2}$$

Note that since the baseline policy $\pi_B$ achieves value 0 in (2), $\zeta$ in Definition 2 is always non-negative. Therefore, any solution $\pi_S$ of (2) is *safe*, because under the true transition probability $P^\star \in \Xi(\widehat{P}, e)$, we have the guarantee that

$$\rho(\pi, P^\star) - \rho(\pi_B, P^\star) \ge \min_{\xi \in \Xi} \left( \rho(\pi, \xi) - \rho(\pi_B, \xi) \right) \ge 0 .$$

It is important to highlight how Definition 2 differs from the standard approach (e.g., [Thomas *et al.*, 2015a]) on determining whether a policy $\pi$ is an improvement over the baseline policy $\pi_B$. The standard approach considers a statistical error bound that translates to the test: $\min_{\xi \in \Xi} \rho(\pi, \xi) \ge \max_{\xi \in \Xi} \rho(\pi_B, \xi)$. The uncertainty parameters $\xi$ on both sides of (2) are not necessarily the same. Therefore, any optimization procedure derived based on this test is more conservative than the problem in (2). Indeed when the error function in $\Xi$ is large, even the baseline policy ($\pi = \pi_B$)

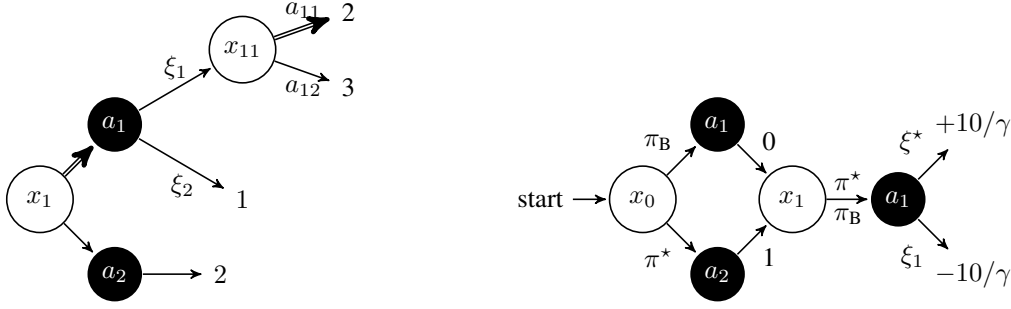

Figure 1: *(left)* A robust/uncertain MDP used in Example 4 that illustrates the sub-optimality of deterministic policies in solving the optimization problem (2). *(right)* A Markov decision process with significant uncertainty in the baseline policy.

may not pass this test. In Section 5.1, we show the conditions under which this approach fails. Our approach also differs from other related work in that we consider regret with respect to the baseline policy, and not the optimal policy, as considered in [Xu and Mannor, 2009].

In the remainder of this section, we highlight some major properties of the optimization problem (2). Specifically, we show that its solution policy may be purely randomized, we compute a bound on the performance loss of its solution policy w.r.t. $\pi^\star$, and we finally prove that it is a NP-hard problem.

## 3.1 Policy Class

The following theorem shows that we should search for the solutions of the optimization problem (2) in the space of randomized policies $\Pi_R$.

**Theorem 3.** *The optimal solution to the optimization problem* (2) *may not be attained by a deterministic policy. Moreover, the loss due to considering deterministic policies cannot be bounded, i.e., there exists no constant $c \in \mathbb{R}$ such that*

$$\max_{\pi \in \Pi_R} \min_{\xi \in \Xi} \Big( \rho(\pi, \xi) - \rho(\pi_B, \xi) \Big) \leq c \cdot \max_{\pi \in \Pi_D} \min_{\xi \in \Xi} \Big( \rho(\pi, \xi) - \rho(\pi_B, \xi) \Big).$$

*Proof.* The proof follows directly from Example 4. The optimal policy in this example is randomized and achieves a guaranteed improvement $\zeta = 1/2$. There is no deterministic policy that guarantees a positive improvement over the baseline policy, which proves the second part of the theorem. $\qquad\square$

**Example 4.** *Consider the robust/uncertain MDP on the left panel of Figure 1 with states $\{x_1, x_{11}\} \subset \mathcal{X}$, actions $\mathcal{A} = \{a_1, a_2, a_{11}, a_{12}\}$, and discount factor $\gamma = 1$. Actions $a_1$ and $a_2$ are shown as solid black nodes. A number with no state represents a terminal state with the corresponding reward. The robust outcomes $\{\xi_1, \xi_2\}$ correspond to the uncertainty set of transition probabilities $\Xi$. The baseline policy $\pi_B$ is deterministic and is denoted by double edges. It can be readily seen from the monotonicity of the Bellman operator that any improved policy $\pi$ will satisfy $\pi(a_{12}|x_{11}) = 1$. Therefore, we will only focus on the policy at state $x_1$. The robust improvement as a function of $\pi(\cdot|x_1)$ and the uncertainties $\{\xi_1, \xi_2\}$ is given as follows:*

$$\min_{\xi \in \Xi} \Big( \rho(\pi, \xi) - \rho(\pi_B, \xi) \Big) = \min_{\xi \in \Xi} \left( \begin{bmatrix} \pi \backslash \xi & \xi_1 & \xi_2 \\ \hline a_1 & 3 & 1 \\ a_2 & 2 & 2 \end{bmatrix} - \begin{bmatrix} \pi \backslash \xi & \xi_1 & \xi_2 \\ \hline a_1 & 2 & 1 \end{bmatrix} \right) = 0.$$

*This shows that no deterministic policy can achieve a positive improvement in this problem. However, a randomized policy $\pi(a_1|x_1) = \pi(a_2|x_1) = 1/2$ returns the maximum improvement $\zeta = 1/2$.*

Randomized policies can do better than their deterministic counterparts, because they allow for hedging among various realizations of the MDP parameters. Example 4 shows a problem such that there exists a realization of the parameters with improvement over the baseline when any deterministic policy is executed. However in this example, there is no single realization of parameters that provides an improvement for all the deterministic policies *simultaneously*. Therefore, randomizing the policy guarantees an improvement independent of the parameters' choice.

## 3.2 Performance Bound

Generally, one cannot compute the truly optimal policy $\pi^\star$ using an imprecise model. Nevertheless, it is still crucial to understand how errors in the model translates to a performance loss w.r.t. an optimal policy. The following theorem (proved in Appendix C) provides a bound on the performance loss of any solution $\pi_S$ to the optimization problem (2).

**Theorem 5.** *A solution $\pi_S$ to the optimization problem* (2) *is safe and its performance loss is bounded by the following inequality:*

$$\Phi(\pi_S) \triangleq \rho(\pi^\star, P^\star) - \rho(\pi_S, P^\star) \leq \min\left\{\frac{2\gamma R_{\max}}{(1-\gamma)^2}\left(\|e_{\pi^\star}\|_{1,u^\star_{\pi^\star}} + \|e_{\pi_B}\|_{1,u^\star_{\pi_B}}\right), \Phi(\pi_B)\right\},$$

*where $u^\star_{\pi^\star}$ and $u^\star_{\pi_B}$ are the state occupancy distributions of the optimal and baseline policies in the true MDP $P^\star$. Furthermore, the above bound is tight.*

## 3.3 Computational Complexity

In this section, we analyze the computational complexity of solving the optimization problem (2) and prove that the problem is NP-hard. In particular, we proceed by showing that the following sub-problem of (2):

$$\arg\min_{\xi \in \Xi}\left(\rho(\pi, \xi) - \rho(\pi_B, \xi)\right), \tag{3}$$

for a fixed $\pi \in \Pi_R$, is NP-hard. The optimization problem (3) can be interpreted as computing a policy that simultaneously minimizes the returns of two MDPs, whose transitions induced by policies $\pi$ and $\pi_B$. The proof of Theorem 6 is given in Appendix D.

**Theorem 6.** *Both optimization problems* (2) *and* (3) *are NP-hard.*

Although the optimization problem (2) is NP-hard in general, but it can be tractable in certain settings. One such setting is when the Markov chain induced by the baseline policy is known precisely, as the following proposition states. See Appendix E for the proof.

**Proposition 7.** *Assume that for each $x \in \mathcal{X}$, the error function induced by the baseline policy is zero, i.e., $e(x, \pi_B(x)) = 0$.[2] Then, the optimization problem* (2) *is equivalent to the following robust MDP (RMDP) problem and can be solved in polynomial time:*

$$\arg\max_{\pi \in \Pi_R}\min_{\xi \in \Xi} \rho(\pi, \xi). \tag{4}$$

## 3.4 Approximate Algorithm

Solving for the optimal solution of (2) may not be possible in practice, since the problem is NP hard. In this section, we propose a simple and practical approximate algorithm. The empirical results of Section 5 indicate that this algorithm holds promise and also suggest that the approach may be a good starting point for building better approximate algorithms in the future.

---

**Algorithm 1:** Approximate Robust Baseline Regret Minimization Algorithm

> **input** : Empirical transition probabilities: $\widehat{P}$, baseline policy $\pi_B$, and the error function $e$
> **output** : Policy $\tilde{\pi}_S$
>
> **1 foreach** $x \in \mathcal{X}, a \in \mathcal{A}$ **do**
>
> **2** $\quad \tilde{e}(x, a) \leftarrow \begin{cases} e(x, a) & \text{when } \pi_B(x) \neq a \\ 0 & \text{otherwise} \end{cases}$ ;
>
> **3 end**
>
> **4** $\tilde{\pi}_S \leftarrow \arg\max_{\pi \in \Pi_R} \min_{\xi \in \Xi(\widehat{P}, \tilde{e})} \left(\rho(\pi, \xi) - \rho(\pi_B, \xi)\right)$ ;
>
> **5 return** $\tilde{\pi}_S$

---

Algorithm 1 contains the pseudocode of the proposed approximate method. The main idea is to use a modified uncertainty model by assuming no error in transition probabilities of the baseline

policy. Then it is possible to minimize the robust baseline regret in polynomial time as suggested by Theorem 7. Assuming no error in baseline transition probabilities is reasonable because of two main reasons. First, in practice, data is often generated by executing the baseline policy, and thus, we may have enough data for a good approximation of the baseline's transition probabilities: $\forall x \in \mathcal{X}, \widehat{P}\big(\cdot\,|x,\pi_B(x)\big) \approx P^\star\big(\cdot\,|x,\pi_B(x)\big)$. Second, transition probabilities often affect baseline and improved policies similarly, and as a result, have little effect on the difference between their returns (i.e., the regret). See Section 5.1 for an example of such behavior.

# 4  Standard Policy Improvement Methods

In Section 3, we showed that finding an exact solution to the optimization problem (2) is computationally expensive and proposed an approximate algorithm. In this section, we describe and analyze two standard methods for computing safe policies and show how they can be interpreted as an approximation of our proposed baseline regret minimization. Due to space limitations, we describe another method, called reward-adjusted MDP, in Appendix H, but report its performance in Section 5.

## 4.1  Solving the Simulator

The simplest solution to (2) is to assume that our *simulator* is accurate and to solve the reward maximization problem of an MDP with the transition probability $\widehat{P}$, i.e., $\pi_{\mathrm{sim}} \in \arg\max_{\pi \in \Pi_R} \rho(\pi, \widehat{P})$. Theorem 8 quantifies the performance loss of the resulted policy $\pi_{\mathrm{sim}}$.

**Theorem 8.** *Let $\pi_{sim}$ be an optimal policy of the reward maximization problem of an MDP with transition probability $\widehat{P}$. Then under Assumption 1, the performance loss of $\pi_{sim}$ is bounded by*

$$\Phi(\pi_{sim}) \triangleq \rho(\pi^\star, P^\star) - \rho(\pi_{sim}, P^\star) \leq \frac{2\gamma R_{\max}}{(1-\gamma)^2}\|e\|_\infty.$$

The proof is available in Appendix F. Note that there is no guarantee that $\pi_{\mathrm{sim}}$ is *safe*, and thus, deploying it may lead to undesirable outcomes due to model uncertainties. Moreover, the performance guarantee of $\pi_{\mathrm{sim}}$, reported in Theorem 8, is weaker than that in Theorem 5 due to the $L_\infty$ norm.

## 4.2  Solving Robust MDP

Another standard solution to the problem in (2) is based on solving the RMDP problem (4). We prove that the policy returned by this algorithm is *safe* and has better (sharper) worst-case guarantees than the simulator-based policy $\pi_{\mathrm{sim}}$. Details of this algorithm are summarized in Algorithm 2. The algorithm first constructs and solves an RMDP. It then returns the solution policy if its worst-case performance over the uncertainty set is better than the robust performance $\max_{\xi \in \Xi} \rho(\pi_B, \xi)$, and it returns the baseline policy $\pi_B$, otherwise.

---
**Algorithm 2:** RMDP-based Algorithm

---

    **input** : Simulated MDP $\widehat{P}$, baseline policy $\pi_B$, and the error function $e$
    **output** : Policy $\pi_{\mathrm{R}}$
**1** $\pi_0 \leftarrow \arg\max_{\pi \in \Pi_R} \min_{\xi \in \Xi(\widehat{P}, e)} \rho(\pi, \xi)$ ;
**2** **if** $\min_{\xi \in \Xi(\widehat{P},e)} \rho(\pi_0, \xi) > \max_{\xi \in \Xi} \rho(\pi_B, \xi)$ **then return** $\pi_0$ **else return** $\pi_{\mathrm{B}}$ ;

---

Algorithm 2 makes use of the following approximation to the solution of (2):

$$\max_{\pi \in \Pi_R} \min_{\xi \in \Xi}\Big(\rho(\pi, \xi) - \rho(\pi_B, \xi)\Big) \geq \max_{\pi \in \Pi_R} \min_{\xi \in \Xi} \rho(\pi, \xi) - \max_{\xi \in \Xi} \rho(\pi_B, \xi),$$

and guarantees safety by designing $\pi$ such that the RHS of this inequality is always non-negative.

The performance bound of $\pi_{\mathrm{R}}$ is identical to that in Theorem 5 and is stated and proved in Theorem 12 in Appendix G. Although the worst-case bounds are the same, we show in Section 5.1 that the performance loss of $\pi_{\mathrm{R}}$ may be worse than that of $\pi_{\mathrm{S}}$ by an arbitrarily large margin.

It is important to discuss the difference between Algorithms 1 and 2. Although both solve an RMDP, they use different uncertainty sets $\Xi$. The uncertainty set used in Algorithm 2 is the true error function in building the simulator, while the uncertainty set used in Algorithm 1 assumes that the error function is zero for all the actions suggested by the baseline policy. As a result, both algorithms approximately solve (2) but approximate the problem in different ways.

## 5 Experimental Evaluation

In this section, we experimentally evaluate the benefits of minimizing the robust baseline regret. First, we demonstrate that solving the problem in (2) may outperform the regular robust formulation by an arbitrarily large margin. Then, in the remainder of the section, we compare the solution quality of Algorithm 1 with simpler methods in more complex and realistic experimental domains. The purpose of our experiments is to show how solution quality depends on the degree of model uncertainties.

### 5.1 An Illustrative Example

Consider the example depicted on the right panel of Figure 1. White nodes represent states and black nodes represent state-action pairs. Labels on the edges originated from states indicate the policy according to which the action is taken; labels on the edges originated from actions denote the rewards and, if necessary, the name of the uncertainty realization. The baseline policy is $\pi_{\mathrm{B}}$, the optimal policy is $\pi^\star$, and the discount factor is $\gamma \in (0, 1)$.

This example represents a setting in which the level of uncertainty varies significantly across the individual states: the transition model is precise in state $x_0$ and uncertain in state $x_1$. The baseline policy $\pi_{\mathrm{B}}$ takes a suboptimal action in state $x_0$ and the optimal action in the uncertain state $x_1$. To prevent being overly conservative in computing a safe policy, one needs to consider that the realization of uncertainty in $x_1$ influences both the baseline and improved policies.

Using the plain robust optimization formulation in Algorithm 2, even the optimal policy $\pi^\star$ is not considered safe in this example. In particular, the robust return of $\pi^\star$ is $\min_\xi \rho(\pi^\star, \xi) = -9$, while the optimistic return of $\pi_{\mathrm{B}}$ is $\max_\xi \rho(\pi_{\mathrm{B}}, \xi) = +10$. On the other hand, solving (2) will return the optimal policy since: $\min_\xi \rho(\pi^\star, \xi) - \rho(\pi_{\mathrm{B}}, \xi) = 11 - 10 = -9 - (-10) = 1$. Even the heuristic method of Section 3.4 will return the optimal policy. Note that since the reward-adjusted formulation (see its description in Appendix H) is even more conservative than the robust formulation, it will also fail to improve on the baseline policy.

### 5.2 Grid Problem

In this section, we use a simple grid problem to compare the solution quality of Algorithm 1 with simpler methods. The grid problem is motivated by modeling customer interactions with an online system. States in the problem represent a two dimensional grid. Columns capture states of interaction with the website and rows capture customer states such as overall satisfaction. Actions can move customers along either dimension with some probability of failure. A more detailed description of this domain is provided in Section I.1.

Our goal is to evaluate how the solution quality of various methods depends on the magnitude of the model error $e$. The model is constructed from samples, and thus, its magnitude of error depends on the number of samples used to build it. We use a uniform random policy to gather samples. Model error function $e$ is then constructed from this simulated data using bounds in Section B. The baseline policy is constructed to be optimal when ignoring the row part of state; see Section I.1 for more details.

All methods are compared in terms of the improvement percentage in total return over the baseline policy. Figure 2 depicts the results as a function of the number of transition samples used in constructing the uncertain model and represents the mean of 40 runs. Methods used in the comparison are as follows: **1)** EXP represents solving the nominal model as described in Section 4.1, **2)** RWA represent the reward-adjusted formulation in Algorithm 3 of Appendix H, **3)** ROB represents the robust method in Algorithm 2, and **4)** RBC represents our approximate solution of Algorithm 1.

Figure 2 shows that Algorithm 1 not only reliably computes policies that are safe, but also significantly improves on the quality of the baseline policy when the model error is large. When the number of

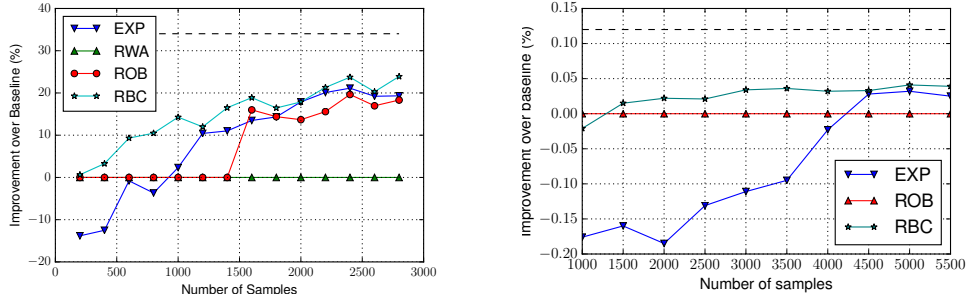

Figure 2: Improvement in return over the baseline policy in: (left) the grid problem and (right) the energy arbitrage problem. The dashed line shows the return of the optimal policy.

samples is small, Algorithm 1 is significantly better than other methods by relying on the baseline policy in states with a large model error and only taking improving actions when the model error is small. Note that EXP can be significantly worse than the baseline policy, especially when the number of samples is small.

## 5.3 Energy Arbitrage

In this section, we compare model-based policy improvement methods using a more complex domain. The problem is to determine an energy arbitrage policy in given limited energy storage (a battery) and stochastic prices. At each time period, the decision-maker observes the available battery charge and a Markov state of energy price, and decides on the amount of energy to purchase or to sell.

The set of states in the energy arbitrage problem consists of three components: current state of charge, current capacity, and a Markov state representing price; the actions represent the amount of energy purchased or sold; the rewards indicate profit/loss in the transactions. We discretize the state of charge and action sets to 10 separate levels. The problem is based on the domain from [Petrik and Wu, 2015], whose description is detailed in Appendix I.2.

Energy arbitrage is a good fit for model-based approaches because it combines known and unknown dynamics. Physics of battery charging and discharging can be modeled with high confidence, while the evolution of energy prices is uncertain. As a result, using an explicit battery model, the only uncertainty is in transition probabilities between the 10 states of the price process instead of the entire 1000 state-action pairs. This significantly reduces the number of samples needed.

As in the previous experiments, we estimate the uncertainty model in a data-driven manner. Notice that the inherent uncertainty is only in price transitions and is independent of the policy used (which controls the storage dynamics). Here the uncertainty set of transition probabilities is estimated using the method in Appendix A, but the uncertainty set is only a non-singleton w.r.t. price states. Figure 2 shows the percentage improvement on the baseline policy averaged over 5 runs. We clearly observe that the heuristic RBC method, described in Section 3.4, effectively interleaves the baseline policy (in states with high level of uncertainty) and an improved policy (in states with low level of uncertainty), and results in the best performance in most cases. Solving a robust MDP with no baseline policy performed similarly to directly solving the simulator.

## 6 Conclusion

In this paper, we study the *model-based* approach to the fundamental problem of learning *safe* policies given a batch of data. A policy is considered *safe*, if it is guaranteed to have an improved performance over a baseline policy. Solving the problem of safety in sequential decision-making can immensely increase the applicability of the existing technology to real-world problems. We show that the standard robust formulation may be overly conservative and formulate a better approach that interleaves an improved policy with the baseline policy, based on the error at each state. We propose and analyze an optimization problem based on this idea (see (2)) and prove that solving it is NP-hard. Furthermore, we propose several approximate solutions and experimentally evaluated their performance.

## Footnotes

[1]From now on, for brevity, we omit the parameters $\widehat{P}$ and $e$, and use $\Xi$ to denote the model uncertainty set.

[2]Note that this is equivalent to precisely knowing the Markov chain induced by the baseline policy $P^\star_{\pi_B}$.

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
