[Supplementary Material · appendix.pdf]

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

## A   Error Bound

Our goal here is to construct the error function $e$, when $\widehat{P}$ is estimated from the samples drawn from $P^\star$, such that we can guarantee that $P^\star \in \Xi(\widehat{P}, e)$, with probability at least $1 - \delta$. Let us assume that at each state-action pair $(x, a) \in \mathcal{X} \times \mathcal{A}$, we draw $N(x, a)$ samples from $P^\star(\cdot | x, a)$.

**Proposition 9.** *If at each state-action pair $(x, a) \in \mathcal{X} \times \mathcal{A}$, we define $e(x, a) = \sqrt{\frac{2}{N(x,a)} \log\left(\frac{|\mathcal{X}||\mathcal{A}|2^{|\mathcal{X}|}}{\delta}\right)}$, then $P^\star \in \Xi(\widehat{P}, e)$, with probability at least $1 - \delta$.*

*Proof.* From Theorem 2.1 in Weissman *et al.* [2003], for each state-action pair $(x, a) \in \mathcal{X} \times \mathcal{A}$, we may write

$$\mathbb{P}\left(||P^\star(\cdot \mid x, a) - \widehat{P}(\cdot \mid x, a)||_1 \geq \epsilon\right) \leq (2^{|\mathcal{X}|} - 2) \exp\left(-\frac{N(x,a)\epsilon^2}{2}\right). \tag{5}$$

Setting $\epsilon = \sqrt{\frac{2}{N(x,a)} \log\left(\frac{|\mathcal{X}||\mathcal{A}|2^{|\mathcal{X}|}}{\delta}\right)}$, we may rewrite (5) as

$$
\begin{aligned}
\mathbb{P}\Big(||P^\star(\cdot \mid x, a) - \widehat{P}(\cdot \mid x, a)||_1 &\geq \sqrt{\frac{2}{N(x,a)} \log\left(\frac{|\mathcal{X}||\mathcal{A}|2^{|\mathcal{X}|}}{\delta}\right)}\Big) \\
&\leq 2^{|\mathcal{X}|} \exp\left(-\frac{N(x,a)}{2} \times \frac{2}{N(x,a)} \log\left(\frac{|\mathcal{X}||\mathcal{A}|2^{|\mathcal{X}|}}{\delta}\right)\right) \\
&= \frac{\delta}{|\mathcal{X}||\mathcal{A}|}. \tag{6}
\end{aligned}
$$

From the definition of the uncertainty set $\Xi(\widehat{P}, e)$ and by summing the error probability in (6), we obtain that $\mathbb{P}\big(P^\star \notin \Xi(\widehat{P}, e)\big) \leq \delta$.   $\square$

# B  Proof of Lemma 11

for which the following technical lemma (whose proof can be found in Appendix B) is used in the analysis.

Before proving Lemma 11, we first prove the following lemma.

**Lemma 10.** *For any policy $\pi \in \Pi_R$, consider two transition probability matrices $P_1$ and $P_2$ and two reward functions $r_1$ and $r_2$ corresponding to $\pi$. Let $v_1$ and $v_2$ be the value functions of the policy $\pi$ given $(P_1, r_1)$ and $(P_2, r_2)$, respectively. Under the assumption that for any state $x \in \mathcal{X}$, $\|P_1(\cdot|x) - P_2(\cdot|x)\|_1 \leq e(x)$, we have*

$$(\mathbf{I} - \gamma P_1)^{-1} \left( r_1 - r_2 - \frac{\gamma R_{\max}}{1 - \gamma} e \right) \leq v_1 - v_2 \leq (\mathbf{I} - \gamma P_1)^{-1} \left( r_1 - r_2 + \frac{\gamma R_{\max}}{1 - \gamma} e \right),$$

*where $e$ is the vector of $e(x)$'s.*

*Proof.* The difference between the two value functions can be written as

$$\begin{aligned}
v_1 - v_2 &= r_1 + \gamma P_1 v_1 - r_2 - \gamma P_2 v_2 \\
&= r_1 + \gamma P_1 v_1 - r_2 - \gamma P_2 v_2 + \gamma P_1 v_2 - \gamma P_1 v_2 \\
&= (r_1 - r_2) + \gamma P_1 (v_1 - v_2) + \gamma (P_1 - P_2) v_2 \\
&= (\mathbf{I} - \gamma P_1)^{-1} \left[ r_1 - r_2 + \gamma (P_1 - P_2) v_2 \right].
\end{aligned}$$

Now using the Holder's inequality, for any $x \in \mathcal{X}$, we have

$$\left| \left( P_1(\cdot|x) - P_2(\cdot|x) \right)^\mathsf{T} v_2 \right| \leq \|P_1(\cdot|x) - P_2(\cdot|x)\|_1 \| v_2 \|_\infty \leq e(x) \| v_2 \|_\infty \leq e(x) \frac{R_{\max}}{1 - \gamma}.$$

The proof follows by uniformly bounding $(P_1 - P_2) v_2$ from the above inequality and from the monotonicity of $(\mathbf{I} - \gamma P_1)^{-1}$. $\square$

**Lemma 11.** *The difference between the returns of a policy $\pi$ in two MDPs parameterized by $P^\star, \xi \in \Xi$ is bounded as*

$$|\rho(\pi, P^\star) - \rho(\pi, \xi)| \leq \frac{2\gamma R_{\max}}{1 - \gamma} p_0^\mathsf{T} (\mathbf{I} - \gamma P_\pi^\star)^{-1} e_\pi,$$

*where $P_\pi^\star$ and $e_\pi$ are the transition probability matrix and error function (between $P^\star$ and $\xi$, see Eq. 1) of policy $\pi$.*

*Proof.* Lemma 11 is the direct consequence of Lemma 10 with the fact that for any $(x, a) \in \mathcal{X} \times \mathcal{A}$ and any $\xi \in \Xi(\widehat{P}, e)$, from Assumption 1 and the construction of $\Xi(\widehat{P}, e)$, we have

$$\begin{aligned}
\|P^\star(\cdot|x, a) - \xi(\cdot|x, a)\|_1 &= \|P^\star(\cdot|x, a) - \widehat{P}(\cdot|x, a) + \widehat{P}(\cdot|x, a) - \xi(\cdot|x, a)\|_1 \\
&\leq \|P^\star(\cdot|x, a) - \widehat{P}(\cdot|x, a)\|_1 + \|\widehat{P}(\cdot|x, a) - \xi(\cdot|x, a)\|_1 \\
&\leq 2e(x, a) .
\end{aligned}$$

$\square$

# C Proof of Theorem 5

To prove the safety of $\pi_S$, note that the objective in (2) is always non-negative, since the baseline policy $\pi_B$ is a feasible solution. Thus, we obtain the safety condition by simple algebraic manipulation as follows:

$$\rho(\pi_S, P^\star) - \rho(\pi_B, P^\star) \geq \min_{\xi \in \Xi} \Big( \rho(\pi_S, \xi) - \rho(\pi_B, \xi) \Big) = \max_{\pi \in \Pi_R} \min_{\xi \in \Xi} \Big( \rho(\pi, \xi) - \rho(\pi_B, \xi) \Big) \geq 0 . \quad (7)$$

Now we prove the performance bound. From Theorem 11, for any policy $\pi$, we may write

$$\max_\xi \Big| \rho(\pi, \xi) - \rho(\pi, P^\star) \Big| \leq \frac{2\gamma R_{\max}}{1-\gamma} p_0^{\mathsf{T}} (\mathbf{I} - P_\pi^\star)^{-1} e_\pi = \frac{2\gamma R_{\max}}{(1-\gamma)^2} \|e_\pi\|_{1, u_\pi^\star} , \quad (8)$$

where $u_\pi^\star$ is state occupancy distribution of policy $\pi$ in the true MDP $P^\star$, defined as

$$u_\pi^\star = (1-\gamma)(\mathbf{I} - \gamma P_\pi^{\star\top})^{-1} p_0.$$

We are now ready to show a bound on the performance loss of $\pi_S$ through the following set of inequalities:

$$\begin{aligned}
\Phi(\pi_S) = \rho(\pi^\star, P^\star) - \rho(\pi_S, P^\star) &= \rho(\pi^\star, P^\star) - \rho(\pi_S, P^\star) + \rho(\pi_B, P^\star) - \rho(\pi_B, P^\star) \\
&\leq \rho(\pi^\star, P^\star) - \rho(\pi_B, P^\star) - \min_\xi \Big( \rho(\pi_S, \xi) - \rho(\pi_B, \xi) \Big) \\
&\leq \rho(\pi^\star, P^\star) - \rho(\pi_B, P^\star) - \min_\xi \Big( \rho(\pi^\star, \xi) - \rho(\pi_B, \xi) \Big) \\
&\leq \rho(\pi^\star, P^\star) - \rho(\pi_B, P^\star) - \min_\xi \rho(\pi^\star, \xi) + \max_\xi \rho(\pi_B, \xi) \\
&= \max_\xi \Big( \rho(\pi^\star, P^\star) - \rho(\pi^\star, \xi) \Big) + \max_\xi \Big( \rho(\pi_B, \xi) - \rho(\pi_B, P^\star) \Big) \\
&\stackrel{(a)}{\leq} \frac{2\gamma R_{\max}}{(1-\gamma)^2} \Big( \|e_{\pi^\star}\|_{1, u_{\pi^\star}^\star} + \|e_{\pi_B}\|_{1, u_{\pi_B}^\star} \Big) ,
\end{aligned} \quad (9)$$

where **(a)** is by applying (8) to the two $\max$ terms on the RHS of the inequality.

The final bound is obtained by combining (9) and the fact that $\rho(\pi_S, P^\star) \geq \rho(\pi_B, P^\star)$, and as a result, $\Phi(\pi_S) \leq \Phi(\pi_B)$.

To prove the tightness of the bound, we use the example depicted in Figure 3. The initial state is $x_0$, actions are $a_1, a_2$, the transitions are deterministic, and the leafs represent absorbing states with the given return. We denote by $P^\star$, the transitions of the true MDP, and by $\xi_1$, the worst-case transitions in the uncertainty set $\Xi(\widehat{P}, e)$. Finally the baseline policy $\pi_B$ takes action $a_1$ in state $x_0$ and shown by double edges in Figure 3. It is clear that the optimal policy $\pi^\star$ is the one that takes action $a_2$ in state $x_0$. The return of this policy is $\rho(\pi^\star, P^\star) = 1 + 2\epsilon$. It is also straightforward to derive that the policy $\pi_S$ that takes action $a_1$ in state $x_0$ (as shown in Figure 3) is a solution to (2). The return of this policy is $\rho(\pi_S, P^\star) = 1$ and its performance loss is $\Phi(\pi_S) = \rho(\pi^\star, P^\star) - \rho(\pi_S, P^\star) = 2\epsilon$.

Figure 3: Example showing the tightness of the bound in Theorem 5.

Now let us set $\epsilon$ in the leafs of Figure 3 to $\epsilon = \frac{2\gamma R_{\max}}{(1-\gamma)^2} \|e_{\pi^\star}\|_{1, u_{\pi^\star}^\star}$. Note that this is the value given by (8) for $\pi = \pi^\star$. This gives us the tightness proof assuming that $\widehat{P}$ is such that $\|e_{\pi^\star}\|_{1, u_{\pi^\star}^\star}$ and $\|e_{\pi_B}\|_{1, u_{\pi_B}^\star}$ have similar values, and $1 + 2\epsilon$ is a valid return value, i.e., $1 + 2\epsilon \leq \frac{R_{\max}}{1-\gamma}$.

Figure 4: MDP $\mathcal{M}_1$ in Theorem 6 that represents the optimization of $\rho(\pi, \xi)$ over $\xi$.

Figure 5: MDP $\mathcal{M}_2$ in Theorem 6 representing the optimization of $\rho(\pi_\mathrm{B}, \xi)$ over $\xi$.

## D  Proof of Theorem 6

Assume a given fixed policy $\pi$. We start by showing the NP hardness of solving (3):

$$\min_{\xi \in \Xi} \left( \rho(\pi, \xi) - \rho(\pi_\mathrm{B}, \xi) \right)$$

by a reduction from the boolean satisfiability (SAT) problem. To simplify the exposition, we also illustrate the reduction on the following simple example SAT problem in a conjunctive normal form (CNF):

$$(a \vee b \vee \neg c) \wedge (\neg a \vee d \vee b) = (l_{11} \vee l_{12} \vee l_{13}) \wedge (l_{21} \vee l_{22} \vee l_{23}) , \tag{10}$$

where $a$, $b$, $c$, and $d$ are the variables, and $l_{ij}$ represent the $j$-th literal in $i$-th disjunction.

As noted above, $\rho(\pi, \xi)$ represents the return of a robust MDP. Recall that computing $\min_\xi \rho(\pi, \xi)$ for a fixed $\pi$ is equivalent to computing a policy in a regular MDP with actions representing realizations of the transition uncertainty. Therefore, optimizing for $\xi$ in (3) translates to finding a single policy $\xi$ for two MDPs—defined by $\pi$ and $\pi_\mathrm{B}$—that maximizes the difference between their returns $\rho(\pi, \xi) - \rho(\pi_\mathrm{B}, \xi)$.

We reduce the SAT problem to the optimization over $\xi$ in (3). As described above, the value $\rho(\pi, \xi)$ for a fixed $\pi$ can be represented as a return of some MDP $\mathcal{M}_1$ for a policy given by $\xi$. Similarly, the value $\rho(\pi_\mathrm{B}, \xi)$ for a fixed $\pi_\mathrm{B}$ can be represented as a return of another MDP $\mathcal{M}2$. We describe the general reduction in detail below. Figures 4 and 5 illustrate the MDPs $\mathcal{M}_1$ and $\mathcal{M}_2$ respectively for the example in (10).

MDPs $\mathcal{M}_1, \mathcal{M}_2$ share the same state and action sets. The actions represent the realization of uncertainty $\xi$ and are denoted by the edge labels. They are discrete and stand for the extreme points of feasible $\ell_1$ uncertainty sets. For ease of notation, we assume $\gamma = 1$ and states with double circles are terminal with rewards inscribed therein. All non-terminal transition have zero rewards.

The identical state set of both $\mathcal{M}_1$ and $\mathcal{M}_2$ are constructed as follows. There is one state for each variable $\mathrm{v} \in \{a, b, c, d\}$, and two states $\{l_{ij}^T, l_{ij}^F\}$ for every literal $l_{ij}$. Informally, actions $\{T, F\}$ for a variable state capture the value of that variable. Actions $\{0, 1\}$ for a literal state $l_{ij}^T$ or $l_{ij}^F$ represent

the value of the variable referenced by the literal. This is regardless of whether the literal is positive or negative. For example, when the variable in $l_{ij}$ is true, the action in $l_{ij}^T$ is 1 and when the variable in $l_{ij}$ is false, the action in $l_{ij}^F$ is 1. Two states per each literal are necessary in order to model the negation operation.

The transitions in MDPs $\mathcal{M}_1$ and $\mathcal{M}_2$ are constructed to guarantee that their returns are $-1$ and $+1$, respectively (and as a result the objective in (3) is $-2$), only if the assignment to the literals satisfies the SAT problem. Note that the transitions for the negated literals, such as $l_{21}$ are different from the positive literals, such as $l_{11}$. This construction easily generalizes to any SAT problem in the CNF. Consider the example in (10) and let $b = T$ (other variables can take any values). It can then be seen readily that the objective in (3) would be $-2$.

Let $\rho^\star$ be the optimal value of (3). Then, to show the correctness of our reduction, we argue that $\rho^\star = -2$, if and only if the SAT problem is satisfiable. To show the reverse implication, assume that the SAT is satisfied for some assignment to variables and construct a policy $\bar{\xi}$ as follows:

$$\bar{\xi}(\text{v}) = \begin{cases} T & \text{if v} = \text{true} \\ F & \text{otherwise} \end{cases}, \qquad \bar{\xi}(l_{ij}^T) = \begin{cases} 1 & \text{if v}_{ij} = \text{true} \\ 0 & \text{otherwise} \end{cases}, \qquad \bar{\xi}(l_{ij}^F) = \begin{cases} 0 & \text{if v}_{ij} = \text{true} \\ 1 & \text{otherwise} \end{cases},$$

where $\text{v}_{ij}$ represents the value of the variable referenced by the corresponding literal $l_{ij}$, e.g., $\text{v}_{11} = \text{v}_{21} = a$ in (10). It can be readily seen that $\rho(\pi, \bar{\xi}) = 1$ and $\rho(\pi_\text{B}, \bar{\xi}) = -1$, and thus, the implication that $\rho^\star = -2$ holds.

To show the forward implication, assume that for an optimal deterministic realization $\bar{\xi}$, we have $\rho(\pi_\text{B}, \xi^\star) = 1$ and $\rho(\pi_\text{B}, \xi^\star) = -1$, and thus, $\rho^\star = -2$. We assign values to variables v as follows:

$$\text{v} = \begin{cases} \text{true} & \text{if } \bar{\xi}(\text{v}) = T \,, \\ \text{false} & \text{otherwise} \,. \end{cases}$$

We have that $\rho(\pi_\text{B}, \bar{\xi}) = 1$ only if for every disjunction $i$ either **1)** there exists a positive literal $l_{ij}$ such that $\bar{\xi}(l_{ij}^T) = 1$ and $\bar{\xi}(l_{ij}^F) = 0$, or **2)** there exists a negative literal $l_{ij}$ such that $\bar{\xi}(l_{ij}^T) = 0$ and $\bar{\xi}(l_{ij}^F) = 1$. Assume without loss of generality that this is always the first literal $l_{i1}$. Now, consider any positive $l_{i1} = \text{v}$ and observe that $\bar{\xi}(l_{i1}^T) = 1$ and $\bar{\xi}(l_{i1}^F) = 0$. Because $\rho(\pi_\text{B}, \bar{\xi}) = 1$ only if $\bar{\xi}(\text{v}) = T$, the disjunction $i$ is satisfied. The case of a negative $l_{i1}$ is analogous, and thus, the forward implication also holds.

The restriction to deterministic policies $\bar{\xi}$ in the forward implication argument can be lifted by considering a discount factor; in such case the maximal return in $\mathcal{M}_2$ may be achieved only by a deterministic policy. Then, appropriately increasing the return in $\mathcal{M}_2$ finishes the argument.

The argument above shows that the inner minimization problem in (2) is NP hard. Recall that (2) is stated as follows:

$$\max_{\pi \in \Pi_R} \min_{\xi \in \Xi} \big( \rho(\pi, \xi) - \rho(\pi_\text{B}, \xi) \big)$$

To prove the theorem, it simply remains to show that the outer maximization over $\pi$ does not make the problem any easier. To show this, we will construct a single robust MDP $\mathcal{R}$ such that a policy with the maximal improvement induces $\mathcal{M}_1$ as the robust optimization subproblem. Baseline policy $\pi_\text{B}$ in $\mathcal{R}$ similarly induces $\mathcal{M}_2$. Then, the difference between improved and baseline policies is no greater than some threshold if and only if the SAT is satisfiable.

Construct the robust MDP $\mathcal{R}$ with the same state set as $\mathcal{M}_1$ and $\mathcal{M}_2$. There are two actions $a_1$ and $a_2$ in each state. Upon taking action $a_1$, the transitions are chosen according to $\mathcal{M}_1$ and the reward is as in $\mathcal{M}_1$. Upon taking action $a_2$, the transition and reward are given the same as in $\mathcal{M}_2$ minus 3. Rewards in terminal states are not modified.

The baseline policy takes action $a_2$, i.e. $\pi_\text{B}(x) = a_2$. Return of the baseline policy is in $[3\,k, 3\,k+1]$ where $k$ is the sum of the number of distinct variables and literals in the CNF.

Let the improvement policy $\pi'$ be $\pi'(x) = a_1$. It can be readily seen that this policy achieves the maximal improvement. This is because $\rho(\pi', \xi) \in [0, -1]$ while the return of any other policy will be at most $-3$ (the return for $a_2$ in any state is $-3$).

To finish the proof, observe that when the SAT is satisfiable then:

$$\max_{\pi \in \Pi_R} \min_{\xi \in \Xi} \big( \rho(\pi, \xi) - \rho(\pi_\text{B}, \xi) \big) = \min_{\xi \in \Xi} \big( \rho(\pi', \xi) - \rho(\pi_\text{B}, \xi) \big) = 3\,k - 2 \,.$$

This is true using the above argument concerning the optimal value of the inner minimization problem. On the other hand, when the SAT is unsatisfiable then by the same argument:

$$\max_{\pi \in \Pi_R} \min_{\xi \in \Xi} \left( \rho(\pi, \xi) - \rho(\pi_{\mathrm{B}}, \xi) \right) = \min_{\xi \in \Xi} \left( \rho(\pi', \xi) - \rho(\pi_{\mathrm{B}}, \xi) \right) > 3\,k - 2\,.$$

This shows that deciding whether the optimal value of (2) is greater than $3\,k - 2$ is as hard as solving the corresponding SAT.

# E  Proof of Proposition 7

The hypothesis in the proposition implies that for any $\xi \in \Xi(\widehat{P}, e)$, we have $\xi(\,\cdot\,|x, \pi_B(x)) = \widehat{P}(\,\cdot\,|x, \pi_B(x))$, $\forall x \in \mathcal{X}$. This further indicates that $\rho(\pi_B, \xi)$ is a constant (independent of $\xi$), for all $\xi \in \Xi(\widehat{P}, e)$. Thus, when the Markov chain induced by the baseline policy is known, the optimization problem (2) is reduced to the optimization problem (4), which is a robust MDP (RMDP) problem with $\ell_1$-constraint uncertainty set. It is known that this class of RMDP problems can be solved in (strongly) polynomial time [Hansen *et al.*, 2013] and has also been solved efficiently in practice [Petrik and Subramanian, 2014].

# F  Proof of Theorem 8

From Lemma 11 with $\pi = \pi_{\text{sim}}$ and $\xi = \widehat{P}$ we have

$$\rho(\pi_{\text{sim}}, \widehat{P}) - \frac{\gamma R_{\max}}{1 - \gamma} p_0^\top (\mathbf{I} - \gamma P_{\pi_{\text{sim}}}^\star)^{-1} e_{\pi_{\text{sim}}} \leq \rho(\pi_{\text{sim}}, P^\star).$$

Thus, we may write

$$\Phi(\pi_{\text{sim}}) \triangleq \rho(\pi^\star, P^\star) - \rho(\pi_{\text{sim}}, P^\star) \leq \rho(\pi^\star, P^\star) - \rho(\pi_{\text{sim}}, \widehat{P}) + \frac{\gamma R_{\max}}{1 - \gamma} p_0^\top (\mathbf{I} - \gamma P_{\pi_{\text{sim}}}^\star)^{-1} e_{\pi_{\text{sim}}}$$

$$\overset{(a)}{\leq} \rho(\pi^\star, P^\star) - \rho(\pi^\star, \widehat{P}) + \frac{\gamma R_{\max}}{1 - \gamma} p_0^\top (\mathbf{I} - \gamma P_{\pi_{\text{sim}}}^\star)^{-1} e_{\pi_{\text{sim}}}$$

$$\overset{(b)}{\leq} \frac{\gamma R_{\max}}{1 - \gamma} p_0^\top \left[ (\mathbf{I} - \gamma P_{\pi^\star}^\star)^{-1} e_{\pi^\star} + (\mathbf{I} - \gamma P_{\pi_{\text{sim}}}^\star)^{-1} e_{\pi_{\text{sim}}} \right]$$

$$\overset{(c)}{\leq} \frac{2\gamma R_{\max}}{(1 - \gamma)^2} \|e\|_\infty \,,$$

where each step follows because:

(a) Optimality of $\pi_{\text{sim}}$ in the MDP with transition probabilities $\widehat{P}$.

(b) Application of Lemma 11 with policy $\pi = \pi^\star$ and $\xi = \widehat{P}$.

(c) For any policy $\pi \in \Pi_R$, we have that $\|p_0^\top (\mathbf{I} - \gamma P_\pi^\star)^{-1}\|_1 = 1/(1 - \gamma)$, and from the application of the Holder's inequality.

# G    Performance Bound on the Solution of the Robust Algorithm

**Theorem 12.** *Given Assumption 1, the nonempty solution $\pi_R$ of Algorithm 2 is safe, i.e., $\rho(\pi_R, P^\star) \geq \rho(\pi_B, P^\star)$. Moreover, its performance loss $\Phi(\pi_R)$ satisfies*

$$\Phi(\pi_R) \triangleq \rho(\pi^\star, P^\star) - \rho(\pi_R, P^\star) \leq \min\left\{ \frac{2\gamma R_{\max}}{(1-\gamma)^2} \left( \|e_{\pi^\star}\|_{1, u_{\pi^\star}^\star} + \|e_{\pi_B}\|_{1, u_{\pi_B}^\star} \right), \Phi(\pi_B) \right\},$$

*where $u_{\pi^\star}^\star$ is the state occupancy distribution of the optimal policy $\pi^\star$ in the true MDP $P^\star$, and $\Phi(\pi_B) = \rho(\pi^\star, P^\star) - \rho(\pi_B, P^\star)$ is the performance loss of the baseline policy.*

*Proof.* To prove the safety of $\pi_R$ and bound its performance loss, we need to upper and lower bound the difference between the performance of any policy $\pi$ in the true MDP $P^\star$ and its worst-case performance in the uncertainty set $\Xi$, i.e., $\min_{\xi \in \Xi} \rho(\pi, \xi)$. Since from Assumption 1, we have $P^\star \in \Xi$, we may write

$$\min_{\xi \in \Xi} \rho(\pi, \xi) \leq \rho(\pi, P^\star). \tag{11}$$

Now let $\bar{\xi} \in \Xi(\widehat{P}, e)$ be the minimizer in $\min_{\xi \in \Xi} \rho(\pi, \xi)$. The minimizer exists because of the continuity and compactness of the uncertainty set. Applying Lemma 11 with $\xi = \bar{\xi}$, for any policy $\pi \in \Pi_R$, we obtain

$$\rho(\pi, P^\star) - \rho(\pi, \bar{\xi}) = \rho(\pi, P^\star) - \min_{\xi \in \Xi} \rho(\pi, \xi) \leq \frac{2\gamma R_{\max}}{1-\gamma} p_0^\top (\mathbf{I} - \gamma P_\pi^\star)^{-1} e_\pi = \frac{2\gamma R_{\max}}{(1-\gamma)^2} \|e_\pi\|_{1, u_\pi^\star},$$
$$\tag{12}$$

where $u_\pi^\star = (1-\gamma) p_0^\top (\mathbf{I} - \gamma P_\pi^\star)^{-1}$ is the state occupancy distribution of policy $\pi$ in the true MDP $P^\star$.

**To prove the safety of the returned policy $\pi_R$:** Consider the two cases on Line 2 of Algorithm 2. When the condition is satisfied, i.e., $\rho_0 > \max_{\xi \in \Xi} \rho(\pi_B, \xi)$, we have

$$\rho(\pi_B, P^\star) \leq \max_{\xi \in \Xi} \rho(\pi_B, \xi) < \underbrace{\min_{\xi \in \Xi} \rho(\pi_0, \xi)}_{\rho_0} \leq \rho(\pi_0, P^\star),$$

where the last inequality comes from (11), and thus, the policy $\pi_R = \pi_0$ is safe. When the condition is violated, then $\pi_R$ is simply $\pi_B$, which is safe by definition.

**To derive a bound on the performance loss of the returned policy $\pi_R$:** Consider also the two cases on Line 2 of Algorithm 2. When the condition is satisfied, using (11), we have

$$\Phi(\pi_R) \triangleq \rho(\pi^\star, P^\star) - \rho(\pi_R, P^\star) = \rho(\pi^\star, P^\star) - \rho(\pi_0, P^\star) \leq \rho(\pi^\star, P^\star) - \min_{\xi \in \Xi} \rho(\pi_0, \xi),$$

and when the condition is violated, we have

$$\Phi(\pi_R) \triangleq \rho(\pi^\star, P^\star) - \rho(\pi_R, P^\star) = \rho(\pi^\star, P^\star) - \rho(\pi_B, P^\star).$$

Since when the condition is satisfied on Line 2 of Algorithm 2, we have

$$\min_{\xi \in \Xi} \rho(\pi_0, \xi) > \max_{\xi \in \Xi} \rho(\pi_B, \xi)$$

in both cases on Line 2 of Algorithm 2, we may write

$$\Phi(\pi_R) \leq \min \left\{ \rho(\pi^\star, P^\star) - \min_{\xi \in \Xi} \rho(\pi_0, \xi) + \max_{\xi \in \Xi} \rho(\pi_B, \xi) - \rho(\pi_B, P^\star), \overbrace{\rho(\pi^\star, P^\star) - \rho(\pi_B, P^\star)}^{\Phi(\pi_B)} \right\}.$$

The first term in the minimum can be written as

$$\rho(\pi^\star, P^\star) - \min_{\xi \in \Xi} \rho(\pi_0, \xi) + \max_{\xi \in \Xi} \rho(\pi_B, \xi) - \rho(\pi_B, P^\star)$$

$$\overset{(a)}{\leq} \rho(\pi^\star, P^\star) - \min_{\xi \in \Xi} \rho(\pi^\star, \xi) + \max_{\xi \in \Xi} \rho(\pi_B, \xi) - \rho(\pi_B, P^\star)$$

$$\overset{(b)}{\leq} \frac{2\gamma R_{\max}}{(1-\gamma)^2} \|e_{\pi^\star}\|_{1, u_{\pi^\star}^\star} + \frac{2\gamma R_{\max}}{(1-\gamma)^2} \|e_{\pi_B}\|_{1, u_{\pi_B}^\star},$$

where **(a)** follows from $\pi_0$ being the solution to (2), and thus, being the maximizer of $\min_{\xi \in \Xi} \rho(\pi, \xi)$, and **(b)** is from (12) with $\pi = \pi^\star$ and $\pi = \pi_B$. $\qquad \square$

# H Solving the Reward-Adjusted MDP

In this section, we describe and analyze another simple method for computing safe policies that we did not include it in Section 4 due to space limitations, and show how it can be interpreted as an approximation of our proposed baseline regret minimization. This algorithm is based on solving a MDP with the same transition probabilities as the simulator, $\widehat{P}$, and rewards defined as $\widehat{r}(x,a) = r(x,a) - \frac{\gamma R_{\max}}{1-\gamma} e(x,a), \ \forall(x,a) \in \mathcal{X} \times \mathcal{A}$. We call this MDP, *reward-adjusted* (RaMDP), and denote its transition probabilities and rewards by $\widetilde{\xi}$. The unique property of RaMDP is that under Assumption 1, the performance of any policy $\pi$ in RaMDP is a lower-bound on its performance in the true MDP, i.e., $\rho(\pi, \widetilde{\xi}) \leq \rho(\pi, P^\star)$ (see Theorem 14). Furthermore in comparison to the objective function of RMDP, the following proposition shows that $\rho(\pi, \widetilde{\xi})$ is always a lower-bound on $\min_{\xi \in \Xi} \rho(\pi, \xi)$.

**Proposition 13.** *Given Assumption 1, for each policy $\pi$, we have* $\min_{\xi \in \Xi} \rho(\pi, \xi) \geq \rho(\pi, \widetilde{\xi})$.

*Proof.* Let $\bar{\xi} \in \Xi(\widehat{P}, e)$ be the minimizer in $\min_{\xi \in \Xi} \rho(\pi, \xi)$. The minimizer exists because of the continuity and compactness of the uncertainty set. From Lemma 10, for each $\pi$, we may write

$$\rho(\pi, \bar{\xi}) \geq \rho(\pi, \widehat{P}) - \frac{\gamma R_{\max}}{1-\gamma} p_0^\mathsf{T} (\mathbf{I} - \gamma \widehat{P}_\pi)^{-1} e_\pi \overset{(a)}{=} \rho(\pi, \widetilde{\xi}),$$

where **(a)** holds because $\widetilde{\xi}$ differs from $\widehat{P}$ only in its reward function, which is of the form $\widehat{r}_\pi = r_\pi - \frac{\gamma R_{\max}}{1-\gamma} e_\pi$. $\qquad\square$

We conclude based on this proposition that the reward-adjusted method approximates the solution of the optimization problem (2) as

$$\max_{\pi \in \Pi_R} \min_{\xi \in \Xi} \left( \rho(\pi, \xi) - \rho(\pi_B, \xi) \right) \geq \max_{\pi \in \Pi_R} \min_{\xi \in \Xi} \rho(\pi, \xi) - \max_{\xi \in \Xi} \rho(\pi_B, \xi)$$

$$\geq \max_{\pi \in \Pi_R} \rho(\pi, \widetilde{\xi}) - \max_{\xi \in \Xi} \rho(\pi_B, \xi), \qquad (13)$$

and guarantees safety by designing $\pi$ such that the RHS of (13) is always non-negative. Algorithm 3 returns an optimal policy of the RaMDP $\widetilde{\xi}$, when the performance of this policy in $\widetilde{\xi}$ is better than the robust baseline performance $\max_{\xi \in \Xi} \rho(\pi_B, \xi)$, and returns $\pi_B$, otherwise.

---

**Algorithm 3:** RaMDP-based Algorithm

    **input** : Simulated MDP $\widehat{P}$, baseline policy $\pi_B$, and the error function $e$
    **output** : Policy $\pi_{Ra}$
1   $\widehat{r}(x,a) \leftarrow r(x,a) - \frac{\gamma R_{\max}}{1-\gamma} e(x,a)$ ;
2   $\pi_0 \leftarrow \arg\max_{\pi \in \Pi_R} \rho(\pi, \widetilde{\xi})$;      where $\widetilde{\xi} = (\widehat{r}, \widehat{P})$
3   $\rho_0 \leftarrow \rho(\pi_0, \widetilde{\xi})$ ;
4   **if** $\rho_0 > \max_{\xi \in \Xi} \rho(\pi_B, \xi)$ **then** $\pi_{Ra} \leftarrow \pi_0$ **else** $\pi_{Ra} \leftarrow \pi_B$ ;
5   **return** $\pi_{Ra}$

---

Since the performance of any policy in the RaMDP $\widetilde{\xi}$ is a lower-bound on its performance in the true MDP $P^\star$, it is guaranteed that the policy $\pi_{Ra}$ returned by Algorithm 3 performs at least as well as the baseline policy $\pi_B$. Theorem 14 shows that $\pi_{Ra}$ is *safe* and quantifies its performance loss.

**Theorem 14.** *Given Assumption 1, the solution $\pi_{Ra}$ of Algorithm 3 is safe, i.e., $\rho(\pi_{Ra}, P^\star) \geq \rho(\pi_B, P^\star)$. Moreover, its performance loss $\Phi(\pi_{Ra})$ satisfies*

$$\Phi(\pi_{Ra}) \overset{\Delta}{=} \rho(\pi^\star, P^\star) - \rho(\pi_{Ra}, P^\star) \leq \min\left\{ \frac{2\gamma R_{\max}}{(1-\gamma)^2} \left( \|e_{\pi^\star}\|_{1, u_{\pi^\star}^\star} + \|e_{\pi_B}\|_{1, u_{\pi_B}^\star} \right), \Phi(\pi_B) \right\},$$

*where $u_{\pi^\star}^\star$ is the state occupancy distribution of the optimal policy $\pi^\star$ in the true MDP $P^\star$, and $\Phi(\pi_B) = \rho(\pi_{\xi^\star}^\star, P^\star) - \rho(\pi_B, P^\star)$ is the performance loss of the baseline policy.*

*Proof.* To prove the safety of $\pi_{Ra}$ and bound its performance loss, we need to upper and lower bound the difference between the performance of any policy $\pi$ in the true MDP $P^\star$ and its performance in $\widetilde{\xi}$, i.e., $\rho(\pi, P^\star) - \rho(\pi, \widetilde{\xi})$. These upper and lower bounds are obtained by applying Lemma 10 with $P_1 = P^\star$, and $P_2 = \widetilde{\xi}$ as follows:

$$\rho(\pi, P^\star) - \rho(\pi, \widetilde{\xi}) \geq p_0^\mathsf{T}(\mathbf{I} - \gamma P_\pi^\star)^{-1}\left(r_\pi - \widehat{r}_\pi - \frac{\gamma R_{\max}}{1-\gamma}e_\pi\right) \geq 0, \tag{14}$$

where the second inequality in (14) follows from the definition of the adjusted reward function $\widehat{r}$, and the fact that $(\mathbf{I} - \gamma P_\pi^\star)^{-1}$ is monotone and $p_0$ is non-negative. Similarly, the upper-bound may be written as

$$\rho(\pi, P^\star) - \rho(\pi, \widetilde{\xi}) \leq \frac{2\gamma R_{\max}}{1-\gamma}p_0^\mathsf{T}(\mathbf{I} - \gamma P_\pi^\star)e_\pi = \frac{2\gamma R_{\max}}{(1-\gamma)^2}\|e_\pi\|_{1,u_\pi^\star}, \tag{15}$$

where $u_\pi^\star = (1-\gamma)p_0^\mathsf{T}(\mathbf{I} - \gamma P_\pi^\star)^{-1}$ is the state occupancy distribution of policy $\pi$ in the true MDP $P^\star$.

**To prove the safety of the returned policy $\pi_{Ra}$:** Consider the two cases on Line 4 of Algorithm 3. When the condition is satisfied, we have

$$\rho(\pi_B, P^\star) \leq \max_{\xi \in \Xi} \rho(\pi_B, \xi) < \rho(\pi_0, \widetilde{\xi}) \leq \rho(\pi_0, P^\star),$$

where the last inequality comes from (14), and thus, the policy $\pi_{Ra} = \pi_0$ is *safe*. When the condition is violated, then $\pi_{Ra}$ is simply $\pi_B$, which is safe by definition.

**To derive a bound on the performance loss of the returned policy $\pi_{Ra}$:** Consider also the two cases on Line 4 of Algorithm 3. When the condition is satisfied, using (14), we have

$$\Phi(\pi_{Ra}) \overset{\Delta}{=} \rho(\pi^\star, P^\star) - \rho(\pi_{Ra}, P^\star) = \rho(\pi^\star, P^\star) - \rho(\pi_0, P^\star) \leq \rho(\pi^\star, P^\star) - \rho(\pi_0, \widetilde{\xi}),$$

and when the condition is violated, we have

$$\Phi(\pi_{Ra}) \overset{\Delta}{=} \rho(\pi^\star, P^\star) - \rho(\pi_{Ra}, P^\star) = \rho(\pi^\star, P^\star) - \rho(\pi_B, P^\star).$$

Since when the condition is satisfied on Line 4 of Algorithm 3, we have

$$\rho(\pi_0, \widetilde{\xi}) > \max_{\xi \in \Xi} \rho(\pi_B, \xi),$$

in both cases on Line 4 of Algorithm 3, we may write

$$\Phi(\pi_{Ra}) \leq \min\left\{\rho(\pi^\star, P^\star) - \rho(\pi_0, \widetilde{\xi}) + \max_{\xi \in \Xi} \rho(\pi_B, \xi) - \rho(\pi_B, P^\star), \overbrace{\rho(\pi^\star, P^\star) - \rho(\pi_B, P^\star)}^{\Phi(\pi_B)}\right\}.$$

The first term in the minimum may be written as

$$\rho(\pi^\star, P^\star) - \rho(\pi_0, \widetilde{\xi}) + \max_{\xi \in \Xi} \rho(\pi_B, \xi) - \rho(\pi_B, P^\star)$$

$$\overset{(a)}{\leq} \rho(\pi^\star, P^\star) - \rho(\pi^\star, \widetilde{\xi}) + \max_{\xi \in \Xi} \rho(\pi_B, \xi) - \rho(\pi_B, P^\star) \overset{(b)}{\leq} \frac{2\gamma R_{\max}}{(1-\gamma)^2}\|e_{\pi^\star}\|_{1,u_{\pi^\star}^\star} + \frac{2\gamma R_{\max}}{(1-\gamma)^2}\|e_{\pi_B}\|_{1,u_{\pi_B}^\star},$$

where (a) follows from $\pi_0$ being an optimal policy of RaMDP $\widetilde{\xi}$ and (b) is from (15) with $\pi = \pi^\star$ and $\pi = \pi_B$. $\qquad\square$

Theorem 14 indicates that by this simple adjustment in the reward function of the simulator $\widehat{P}$, we may guarantee that our solution is *safe*. Moreover, it shows that the bound on the performance loss of $\pi_{Ra}$ is actually tighter than that for the solution $\pi_{\text{sim}}$ of the simulator, reported in Theorem 8.

While Algorithm 2 is more complex than Algorithm 3 (since solving a RMDP is more complicated than a standard MDP), Theorem 12 does not show any advantage for $\pi_R$ over $\pi_{RA}$, neither in terms of safety nor in terms of the bound on its performance loss (compared to Theorem 14). On the other

hand, while Algorithm 3 guarantees to yield a safe policy more efficiently than Algorithm 2, from Proposition 13 one notices that Algorithm 3 may be overly conservative in many circumstances. This is because the adjustment of the reward function is based on the assumption that there exists a state with the maximum value of $R_{\max}/(1-\gamma)$ and that this state is accessible from all other states with reward $R_{\max}$. Thus, we may conclude that Algorithm 2 returns a less conservative safe policy (compared to Algorithm 3), with extra computational cost.

The experimental results of Section 5 also show that the reward-adjusted solution of Algorithm 3 can be extremely conservative.

# I  Description of Experimental Domains

## I.1  Grid Problem

We now describe the grid problem in more detail. The state space in the problem comes from a two-dimensional grid: $\mathcal{S} = \{s_{ij} \ : \ i \in \mathcal{I}, j \in \mathcal{J}\}$; $i$ and $j$ represent a column and row respectively. Columns represent states of an interaction with the website, and rows represent more complex customer states, such as overall satisfaction. The dimensions are $|\mathcal{I}| = 12$ and $\mathcal{J} = 3$.

There are 4 actions: $a_L$ for left, $a_R$ for right, $a_U$ for up, and $a_U$ for down. Rewards are independent of actions and depend only on states and only on the column: $x_{ij} = r_i$ where $r = [-1, 1, 2, 3, 2, 1, -1, -2, -3, 3, 4, 5]$. Actions left and right generally decrease and increase the column number; but can fail and in that case the transition is to a random column. The failure probability $z_j$ depends on the row $j$, with specific failure probabilities: $z = [0.9, 0.2, 0.3]$. If a transition fails, then the next state is chosen according to a fixed distribution which is generated a priori from a Dirichlet distribution. The distribution for first and last row are the same, and the middle row is the average of the two.

Algorithm 4 describes how the transition from a state is computed. The initial state is $s_{00}$.

---

**Algorithm 4:** Transitions for state and action.

    **Data:** Current state $s_{ij}$, action $a$, distributions $P_j$
    **Result:** Next state $s_{kl}$
**1** **if** *Random uniform from* $[0, 1] > z_j$ **then**
**2**     **if** $a = a_R$ **then**
**3**         $k \leftarrow i + 1$ ;
**4**     **else if** $a = a_L$ **then**
**5**         $k \leftarrow i - 1$ ;
**6**     **else**
**7**         $k \leftarrow$ Random from $P_j$ ;
**8**     **end**
**9**     $k \leftarrow \max\{0, \min\{k, |\mathcal{I}| - 1\}\}$ ;
**10** **else**
**11**     $k \leftarrow$ Random from $P_j$ ;
**12** **end**
**13** **if** $a = a_U$ **then**
**14**     $l \leftarrow j + 1$ ;
**15** **else if** $a = a_D$ **then**
**16**     $l \leftarrow j - 1$ ;
**17** **else**
**18**     $e \leftarrow$ Random uniform from $[0, 1]$ ;
**19**     **if** $e \leq 0.35$ **then**
**20**         $l \leftarrow j + 1$ ;
**21**     **else if** $e \leq 0.7$ **then**
**22**         $l \leftarrow j - 1$ ;
**23**     **else**
**24**         $l \leftarrow j$ ;
**25**     **end**
**26** **end**
**27** $l \leftarrow \max\{0, \min\{l, |\mathcal{J}| - 1\}\}$ ;
**28** **return** $s_{kl}$;

---

## I.2  Energy Arbitrage

The energy arbitrage model is based on [Petrik and Wu, 2015] using a discount factor $0.9999$. We summarize it here for ease of reference. Recall that even though the state and action spaces in this problem are continuous, we discretize them as described in Section 5.

The problem represents an energy arbitrage model with multiple finite *known* price levels and a stochastic evolution given a limited storage capacity. In particular, the storage is assumed to be an electrical battery that degrades when energy is stored or retrieved. Energy prices are governed by a Markov process with states $\Theta$. There are two energy prices in each time step: $p^i : \Theta \to \mathbb{R}_+$ is the purchase (or input) price and $p^o : \Theta \to \mathbb{R}_+$ is the sell (or output) price. Energy prices $\theta$ vary between 0 and 10 and their evolution is governed by a martingale with a normal distribution around the mean.

We use $s$ to denote the available battery capacity with $s_0$ denoting the initial capacity. The current state of charge is denotes as $z$ or $y$ and must satisfy that $0 \le z_t \le s_t$ at any time step $t$. The action is the amount of energy to charge or discharge, which is denoted by $a$. A positive $a$ indicates that energy is purchased to charge the battery; a negative $a$ indicates the sale of energy.

The battery storage degrades with use. The degradation is a function of the battery capacity when charged or discharged. We use a general model of battery degradation with a specific focus on Li-ion batteries. The degradation function $d(z, a) \in \mathbb{R}_+$ represent the battery capacity loss after starting at the state of charge $z \ge 0$ and charging (discharging if negative) by $a$ with $-z \le a \le s_0$. This function indicates the loss of capacity, such that:

$$s_{t+1} = s_t - d(z_t, a_t)$$

The state set in the Markov decision problem is composed of $(z, s, \theta)$ where $z$ is the state of charge, $s$ is the battery capacity, and $\theta \in \Theta$ is the state of the price process. The available actions in a state $(z, s, \theta)$ are $a$ such that $-z \le a \le s - z$. The transition is from $(z_t, s_t, \theta_t)$ to $(z_{t+1}, s_{t+1}, \theta_{t+1})$ given action $a_t$ is:

$$z_{t+1} = z_t + a_t$$
$$s_{t+1} = s_t - d(z_t, a_t)$$

The probability of this transition is given by $P[\theta_{t+1} | \theta_t]$. The reward for this transition is:

$$r((z_t, s_t, \theta_t), a_t) = \begin{cases} -a_t \cdot p^i - c^d \cdot d(z_t, a_t) & \text{if } a_t \ge 0 \\ -a_t \cdot p^o - c^d \cdot d(z_t, a_t) & \text{if } a_t < 0 \end{cases}.$$

That is, the reward captures the monetary value of the transaction minus a penalty for degradation of the battery. Here, $c^d$ represents the cost of a unit of lost battery capacity.

The Bellman optimality equations for this problem are:

$$\begin{aligned} q_T(z, s, \theta) &= 0 \\ v_t(z, s, \theta_t) &= \min\big\{ p^i_{\theta_t} [a]_+ + p^o_{\theta_t} [a]_- + \\ &\quad + c^d\, d(z, a) + \\ &\quad + q_t(z + a, s - d(z, a), \theta_t) : \\ &\quad\quad : a \in [-z, s - z] \big\} \\ q_t(z, s, \theta_t) &= \lambda \cdot \mathrm{E}[v_{t+1}(z, s, \theta_{t+1})] \end{aligned} \tag{16}$$

where $[a]_+ = \max\{a, 0\}$ and $[a]_- = \min\{a, 0\}$ and the expectation is taken over $P(\theta_{t+1} | \theta_t)$.

Please see [Petrik and Wu, 2015] for more details, including the price transition matrix.