[Reviews · NeurIPS 2016]

Reviewer 1

Summary

The paper considers the problem of robust MDP. In particular, it considers that we have an uncertain model of transition probabilities of the MDP (with the rectangular uncertainty) as well as a baseline policy. The goal is to find a new policy such that it is guaranteed to be no worse than the baseline. This is called the safe policy improvement. A robust approach would find a policy that maximizes the worst case performance. To provide a safe policy, it has to compare it with the performance of the baseline. Since the performance of the baseline isn’t known, one can compare it with the best possible performance of the baseline. However this might be an overly pessimistic approach because the choice of the worst model for the robust policy is not necessarily the same as the choice of the best model selected to evaluate the baseline policy. A more reasonable approach is to compare the regret of the policy compared to the baseline. The regret of a policy is defined as the difference of the performance of the baseline policy with the performance of the policy under the same probability model. By finding the policy that maximizes the negative worst case regret within the uncertainty set, we ensure that the resulting policy is safe, and hopefully less conservative than the previous approach. The paper provides a performance bound for this solution (Theorem 5). It also shows that in general such a policy is randomized (Theorem 3). The difficulty, however, is that the corresponding optimization problem is NP-Hard (Theorem 6). To circumvent this challenge, the paper suggests a heuristic approximation to this problem. The approximation is based on the fact that if there is no uncertainty for the MDP when actions are selected according to the baseline policy, the problem becomes a usual robust MDP problem (Proposition 7), which can be solved in polynomial time. The approximation is that the algorithm enforces the uncertainty for the actions selected by the baseline policy to be zero, as if there is no error in their model. This type of approximation is reasonable when the data is collected mostly from following the baseline policy with some occasional exploratory randomness to select other actions. In that case, the estimated model for the baseline is indeed much more accurate. Note that this is still an approximation. The paper does not provide a guarantee on the quality of this approximation, but it empirically shows that the performance is better than a usual robust MDP approach that doesn’t change the uncertainty set (Algorithm 2). It is also much better than a certainty-equivalence approach that does not take the uncertainty into account (Algorithm 1).

Qualitative Assessment

Brief summary of evaluation: Technical quality: The results seem to be technically sound. The technical tools are more or less standard. Novelty/originality: The optimization formulation seems to be novel. The derivation of the error bounds is more or less standard. Potential impact or usefulness: This might be a stepping stone for further results, but I doubt it would have a huge real-world impact as it is, because a) it works only for finite MDPs and b) the solution doesn’t have a guarantee. Clarity and presentation: It is relatively well-written. === This is a reasonably good paper. It addresses an important problem that is of interest to the real-world applications of reinforcement learning. It formulates the problem as an optimization problem that is much more reasonable than a conventional robust MDP formulation. It is unfortunate that the solution to the new optimization problem is NP-Hard (Theorem 6). However showing that this is the case is important. Also providing the theoretical guarantee on how well we can expect the optimal solution to behave is important too (Theorem 5). The suggested algorithm (Algorithm 1) is indeed very simple. It would be much better to have a theoretical guarantee on the performance of the approximate algorithm, but in lieu of that, the empirical results are reassuring. Some detailed comments: - Algorithm 1 (L4): Shouldn’t the term rho(pi_B,xi) be removed? It is constant anyway. - L198: The reason #2 is not clear. - L69: Please specify that the paper is about finite state and action spaces. - L159 (Theorem 5): The norms as well as e_{\pi*} and e_{\pi_B} are not defined. - L90: What does a randomized uncertain set mean? - Please compare the choice of regret as the objective to be minimized with the following work: Regan and Boutilier, “Regret-based Reward Elicitation for Markov Decision Process,” USI, 2009. The paper is not about the uncertainty in P, but they choose regret too. === After Rebuttal: Thank you for the response.

Confidence in this Review

2-Confident (read it all; understood it all reasonably well)


Reviewer 2

Summary

The problem considered is improving a baseline policy when an MDP with a *known* uncertainty about its transitions is available. The authors propose to solve an optimization problem of maximizing the difference in return between the new policy and the baseline, under the worst possible model in the uncertainty set. Since the same model is used both for the new policy and the baseline, this results in a less conservative formulation than a standard robust MDP. However, this problem is shown to be NP-hard, and a heuristic approximation is proposed, and validated empirically.

Qualitative Assessment

I enjoyed reading this paper. The examples throughout the paper are illuminating, and the theoretical results are clearly presented. The proposed optimization problem is adequately investigated, and the comparison to existing algorithms is interesting. Robust MDPs have been around for quite some time, though they are often criticized of being too conservative to be of practical use. I hope this original application, which seems like a natural way to overcome this conservativeness, will lead to wider use of robust MDPs. I have one concern that I'd like the authors to clarify: If I understand correctly, Algorithm 1 comes with no theoretical guarantees of safety, nor performance bounds, due to the approximation involved. In this case, how is it better than simply solving the robust MDP: max_pi min_{\xi \in \Xi(\hat{P}, e)} \rho(\pi,\xi) ? (i.e., Algorithm 2, but without the safety criterion in line 2 of Algorithm 2) I would think that this is a reasonable baseline to compare with, at least empirically, as it measures the significance of the proposed heuristic modification to the uncertainty set. Since it also comes with no guarantees, there is no prior reason to prefer Algorithm 2 over it. Minor comments: 134: This is subjective, but I find it more readable to give examples their own counter. 166: The problem of simultaneously minimizing the returns of two MDPs may be of independent interest. Is it a new result that it is NP-hard? Does your proof cover this general case? If so, I would consider presenting it as such. 190: Proposition 7

Confidence in this Review

2-Confident (read it all; understood it all reasonably well)


Reviewer 3

Summary

This paper studies the problem of designing safe algorithms for reinforcement learning, where safe is defined with respect to a given baseline. The paper proposes a new notion of safety, where we seek the maximin improvement over a given baseline policy. The paper presents a number of results related to this concept, and an approximate algorithm.

Qualitative Assessment

The paper gives a new perspective on the question of safe RL algorithms. The development is sometimes hard to follow, and it would have been nice to review existing notions of safety before delving into the novel idea. Having whole algorithms relegated to the appendix doesn't seem right to me. Issues/questions -- Line 104: Where is that definition of safety from? In particular, Thomas et al.'s "High Confidence Policy Improvement" provides a high-probability, value-based definition. The paper should clarify if/why there are multiple definitions, and why we prefer one over the other. -- Theorem 3: Why is a multiplicative bound from the right notion? Why not an additive bound (which wouldn't produce the same conclusion)? -- Proposition 7: Can you bound the approximation error when your assumption does not hold? In particular, Remark 1: can Algorithm 1 be arbitrarily worse than the exact solution? -- In the proof of Theorem 5. Can you explain the passage on line 392 from rho(pi_S, xi) to rho(pi^*, xi)? It doesn't seem obvious to me, because pi^* is the optimal policy for P^*, not for any xi. -- Proof of NP-hardness: the proof is hard to follow. Can you explain why some states (e.g. l_13) are missing at the bottom of Figure 6? Also, I believe the statement needs to be formulated more carefully, and isn't correct as-is. The uncertainty set needs to be described in terms of its independent factors. I.e., if Xi is provided as one big set to the algorithm, then the minimization is trivially polynomial in the exponentially-sized Xi. Minor issues -- P^* seems superfluous. The MDP is already defined as using P. Why not use P instead of P^*? -- Def. 2: are we looking for a maximal zeta or a max-achieving pi? -- Line 118: "standard approach" sounds definitive and grates on me, especially since the paper at that point had not clearly reviewed what existing safe methods there are. -- Line 160: Your state occupancy measures are wrt p_0, and are discounted -- that should be clarified. -- Theorem 8 and Lemma 11: There's a bit of slop in the use of the e, e_pi vectors. These need a bit of guesswork. It would be good to define the two vectors carefully.

Confidence in this Review

2-Confident (read it all; understood it all reasonably well)


Reviewer 4

Summary

This paper proposes a new formulation for robust optimization that combines the maximization over policies and minimizations over transition models into a single expression. The new objective jointly considers performance of the proposed policy and performance of the baseline policy under a shared (minimized) model. The formulation allows the policy to improve even with significant uncertainties in the transition probabilities.

Qualitative Assessment

The original formulation of robust optimization evaluates the baseline policy under the most optimistic model and the proposed policy under the most pessimistic model. For some domains this is a very punishing formulation because the most pessimistic model may never have an expectation as great as the most optimistic model-- regardless of the number of samples collected to build the model. By considering the baseline policy and the proposed policy jointly, this formulation does not need to wait until the proposed policy provably improves on the baseline policy's optimistic performance. I do not have much experience with robust optimization, so I cannot evaluate this work in the larger context of this subfield, but I find the reformulation to be convincing. My only concern is with the likely additional cost of optimizing to this formulation. Since the authors prove the formulation is NP-hard, their algorithms focus on promising approximations. nitpick: I could be missing something, but I think the assumption that transition probabilities are known for actions consistent with baseline policy is not realistic in the energy market example. This sounds like the action may be controlling the price? I am interested in learning more about the details here.

Confidence in this Review

1-Less confident (might not have understood significant parts)


Reviewer 5

Summary

This paper address the problem of safe policy, whose worst case is guaranteed by a baseline policy. Contrary to the previous approaches which rely on overly conservative formulation, the proposed approach helps finding better policy while guarantee the better discounted return over the baseline policy. As the proposed approach is computationally intractable, the authors proposed a reasonable approximation. The approximation method is theoretically and empirically analysed.

Qualitative Assessment

This is the solid work addressing the limitation of existing work and propose a better approach. Even though the realized method is the approximation of the ideal approach, this paper justify the necessity of the approximation, and empirically shows good performance over other possible approaches. The authors also derive the performance bound of various approach for learning safe policy, but the comparison among derived bounds are less described.

Confidence in this Review

2-Confident (read it all; understood it all reasonably well)